# A Study on the Vibration Characteristics and Damage Mechanism of Pantograph Strips in a Railway Electrification System

Qirui Wu, Xiaohan Phrain Gu *, Ziyan Ma and Anbin Wang

School of Urban Railway Transportation, Shanghai University of Engineering Science, 333 Longteng Road, Shanghai 201620, China

* Correspondence: evxxhpgu@sues.edu.cn

**Abstract:** This paper presents the vibration characteristics of a pantograph–catenary interaction in a rigid catenary system. Both computational simulation and laboratory tests are carried out to evaluate the frequency contents of pantograph strips. Based on the observation that irregular wear is characterized by the consistency between the pantograph strips' wear pattern and the mode shape of their dominant modal frequencies, it is deducted that resonance occurs at the pantograph strip and the contact wire interface in the high frequency range. By applying damping treatment to the pantograph strip, and hence improving its damping property, a reduction of 7 dB in the total vibration level at the sliding contact can be achieved, as verified through field tests. It is also found that the worse the initial condition of the pantograph–catenary system, the more prominent the damping effects on the control of high-frequency vibration for irregular wear problems.

**Keywords:** pantograph strip; vibration characteristics; pantograph–catenary system; rigid catenary; irregular wear; damping performance




## 1. Introduction

### 1.1. Pantograph–Catenary Interaction

In modern railway engineering, especially in urban rail transit systems, such as the metro, the pantograph–catenary system, shown in Figure 1, provides the means of electrification for train vehicles. Pantograph–catenary interaction is one of the two main coupling problems in railway systems, the other being wheel–rail coupling. The interaction between the pantograph and catenary is complex, featuring both mechanical and electrical wear. Good contact quality of the pantograph–catenary system is crucial to power quality, operational safety, and maintenance costs.

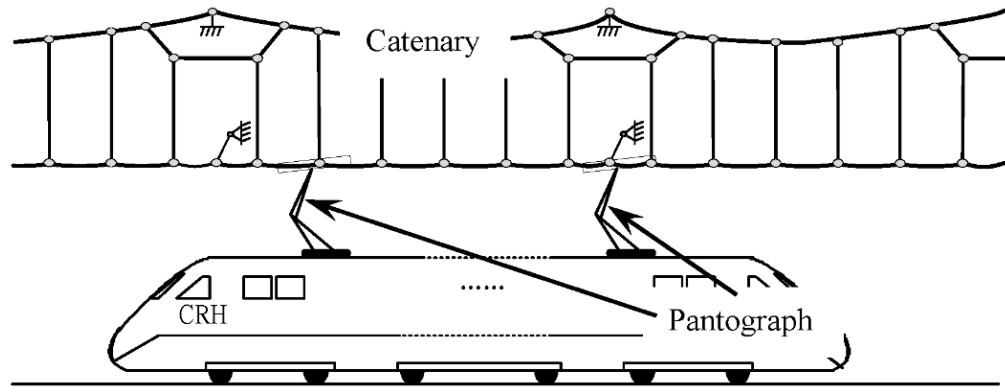

**Figure 1.** The pantograph–catenary system (traditional catenary system with suspension wires).

As early as the 1960s, a simple computational model was set up to study the dynamic behaviour of the pantograph–catenary system by assessing some intrinsic properties of

the system, such as tension in the wire and the damping of both the catenary wire and the pantograph frame [1]. Experimental works using mainly wear machines were conducted for the wear of the material of the contact wire and pantograph sliding strip [2–4]. Different types of wear at the pantograph strip and catenary wire interface were closely examined. A combined adhesive and abrasive wear regime was established, with intensifying effects on the wear rate and the failure at the sliding interface highlighted [5].

As the train speed has increased, more researchers have applied computational simulations to study the interactive system. A simple single degree of freedom model was constructed to examine the dynamic behaviour of the system. It was found that the dynamic stiffness of the catenary varies with train speed [6,7]. Nonlinearity in the coupled system is accounted for by using partial differential equations and differential algebraic equations [8]. A three-dimensional model, allowing lateral wind effects to be taken into consideration, as well as incorporating the pantograph as an articulated multibody system, was developed for a more realistic simulation [9]. A moving mesh algorithm, enabling the contact between moving loads and cable structures, facilitated a much faster computation with an equivalent accuracy [10]. The vibration characteristics of the rigid catenary with different supporting lengths were examined; vibration characteristics of both the catenary system and the pantograph were investigated in a low-frequency range of no more than 15 Hz [11].

Research works have been carried out on the optimization of the dynamic performance of the coupled system. The dynamic behaviour of multiple pantographs on a tilted train, to better represent the reality, was found to be critical to high-speed trains. Based on this, various constructions of passive pantographs and proposals for active control concepts have been introduced [12]. Similarly, active feedback control of the pantograph–catenary system was achieved through evening out the catenary equivalent stiffness variation by making the contact force equivalent to its constant reference [13]. The active control of the pantograph was designed with two different configurations of actuator installation. Pros and cons of the two installation configurations were discussed with regard to the fluctuation of contact force and robustness against the controller time delay [14]. Parametric studies of the stiffness and the damping of the pantograph, as well as the static force and the tension of contact wire were carried out, based on which design parameters could be optimized [15].

The system dynamic behaviour in the medium-high frequency range was evaluated, with the aim of optimizing line design and pantograph service conditions [16]. In this direction, it was found in further studies that the arc intensity was correlated with the vertical acceleration of the contact strip, and that the contact force was influenced by contact wire irregularity [17,18].

Others have looked into some non-mechanical aspects, such as the tribological behaviour of the contacting material [19–21], the effect of arc discharge on wear rate [22], the effect of temperature and arc discharge on friction, and wear behaviour [23].

So far, research works in the field mainly concern: (i) the traditional suspension wire type of catenary, and (ii) system dynamic behaviour in a low-frequency range, i.e., no more than 100 Hz. Few scholars have looked into the pantograph strip and contact wire interface of a traditional suspension wire system; but the vibration effect and contact loss at the sliding interface in the traditional suspension wire system are less prominent than those in the rigid catenary system.

*1.2. Irregular Wear*

Compared to the traditional catenary system with suspension wires, the rigid catenary system, as shown in Figure 2, has advantages such as a greater capacity of current carrying, better reliability, maintainability and availability, and therefore overall lower life-cycle costs. Rigid catenaries have been increasingly used in tunnel sections of metro systems in China, South Korea, and Europe. That said, due to the lack of tension and the discontinuous layout of the aluminium frames, the contact quality of these system is largely compromised.

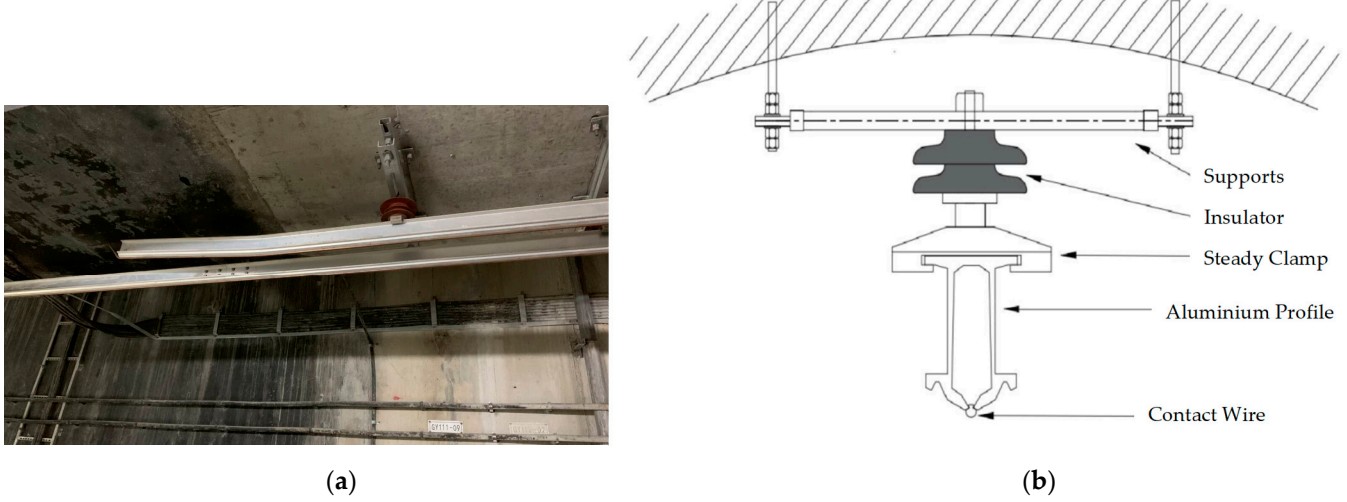

**Figure 2.** A rigid catenary system. (**a**) Discontinuous layout along the rigid catenary. (**b**) The cross-section and key components of a rigid catenary system.

In plan view, the layout of a rigid catenary typically comprises a sine wave or Z-shape in relation to the track centre-line. The catenary wire runs back and forth across the pantograph strip to achieve an even wear transversely across the pantograph strip. As the train vehicle runs at speeds through the catenary, defects, irregular profiles and discontinuities along the rigid catenary wire, as well as external factors, cause high levels of acceleration of the contact force, and even impact loads. This potentially introduces contact loss and vibration at the sliding interface, together with adhesive and abrasive wear, resembling the vibration-assisted drilling process, and hence excessive wear in the system. Both mechanical wear and electrical wear contribute to the overall wear. Loss of contact at the sliding interface intensifies electrical wear, even causes arc events. At this stage of the study, only mechanical wear is considered.

Irregular wear of the pantograph–catenary system has been found in metro lines with rigid catenaries in cities in China.

The design life of a standard pantograph stirp is approximately 100,000 km in China. In the metro system, this corresponds to between 1 and 3 years of service life, depending on the operational condition of a specific metro line, including the condition of the pantograph–category system. Irregular wear, however, is featured with uneven and excessively high wear depths taking place within very short periods of time, e.g., several days or weeks. Figure 3 shows the daily average wear rate of pantograph strips, measured at the deepest trough, on representative rail vehicles of the metro line 1 of S city. Comparing to the normal carbon strip wear rate of 0.9–1.9 mm per 10,000 km, irregular wear increases the pantograph strip wear rate to some 148.2 mm per 10,000 km. Representative pictures of worn conditions of pantograph strips after a few days from new are shown in Figure 4.

In the attempt of addressing the irregular wear problem, practitioners in the industry have tried using various modern technologies, such as AI cameras, high-precision scanning devices, etc., for the health monitoring of the catenary system. Limited research works have also been conducted, for instance, to study the influence of the layout of rigid catenary systems [24]. However, the cause of irregular wear is yet to be found.

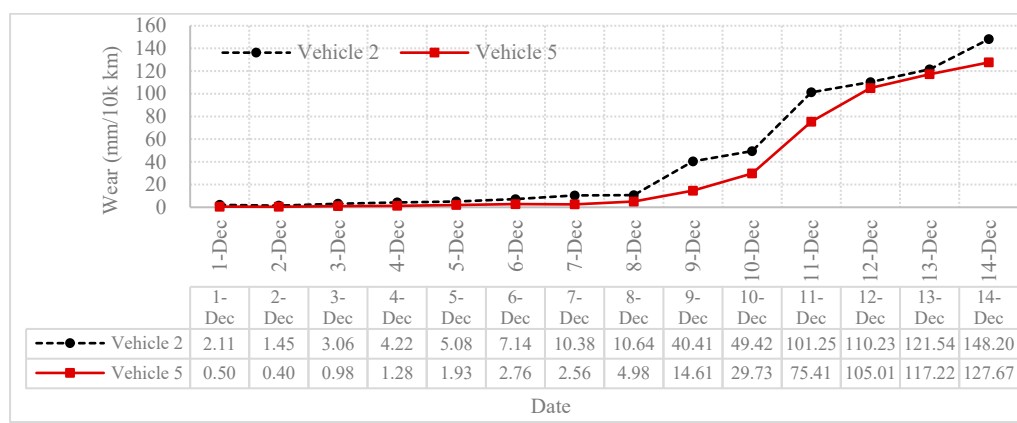

**Figure 3.** Pantograph strip average daily wear rate before and during irregular wear.

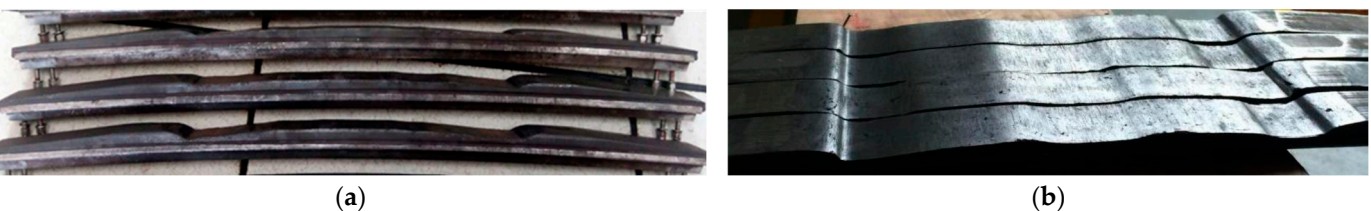

| (a) | (b) |

**Figure 4.** Worn conditions of pantograph strips with irregular wear. (**a**) Pantograph strips with two-groove wear. (**b**) Pantograph strips with three-groove wear.

## 2. Computational Simulation of the Pantograph–Catenary System

### 2.1. The Rigid Catenary Model

Figure 5 below is a schematic of a simplified anchor section of a rigid catenary system. Each anchor section, which is typically 250 m in length, comprises a number of aluminium sections with a typical section length $L$ varying from 6 m to 12 m between two hanging points. Usually, as the train speed increases, aluminium section length $L$ decreases. The model is simplified so that the aluminium profile and contact wire are treated together as a simply supported beam, with two degrees of freedom in the vertical and longitudinal (along the track) directions. The equation of motion of the simplified system is, therefore, shown in Equation (1), with parameters defined in Table 1.

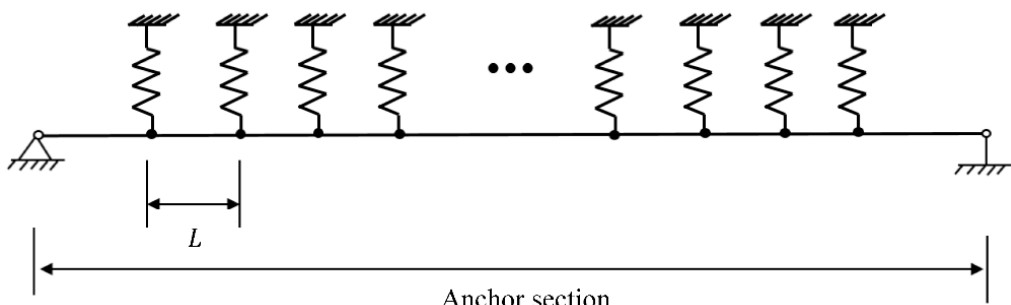

**Figure 5.** A simplified schematic of the layout of the rigid catenary.

Inputting the boundary condition as per Equation (2) and the initial condition as per Equation (3), the harmonics vibration equation as per Equation (4) and the mode shape orthogonality as per Equation (5) all into Equation (1), together with the Rayleigh damping model of Equation (6), the equation of motion of the rigid catenary system can be written as Equation (7) below.

$$\rho A \frac{\partial^2 y(x,t)}{\partial t^2} + EI \frac{\partial^4(x,t)}{\partial x^4} + \sum_{j=1}^{m} m_c \frac{\partial^2 y(x,t)}{\partial t^2} \delta(x - x_j) + \sum_{k=1}^{n} m_z \frac{\partial^2 y(x,t)}{\partial t^2} \delta(x - a_k) + \sum_{j=1}^{m} k_{dc} y(x,t) \delta(x - x_j)$$
$$= f_c(t - t_0)\delta(x - (x_0 + v(t - t_0))) - \rho A g - m_c g \sum_{j=1}^{m} \delta(x - x_j) - m_z g \sum_{k=1}^{n} \delta(x - a_k) \tag{1}$$

$$y(0,t) = 0; \ \frac{\partial^2}{\partial x^2} y(x,t)\big|_{x=0} = 0; y(L,t) = 0; \ \frac{\partial^2}{\partial x^2} y(x,t)\big|_{x=L} = 0. \tag{2}$$

$$y(x,0) = 0; \ \frac{\partial}{\partial t} y(x,t)\big|_{t=0} = 0 \tag{3}$$

$$\varnothing_i(x) = A_1 \cos(ax) + A_2 \sin(ax) + A_3 \cosh(ax) + A_4 \sinh(ax) \tag{4}$$

$$y(x,t) = \sum_{i=1}^{N} \varnothing_i(x) q_i(t); \ \varnothing_i(x) = \sqrt{\frac{2}{\rho A L}} \sin\left(\frac{i\pi x}{L}\right) \tag{5}$$

$$C_\varnothing = \alpha M_\varnothing + \beta K_\varnothing \tag{6}$$

$$M_\varnothing \ddot{q}_\varnothing + C_\varnothing \dot{q}_\varnothing + K_\varnothing q_\varnothing = G_\varnothing + P_{mc\varnothing} + P_{mz\varnothing} + F_{c\varnothing} \tag{7}$$

**Table 1.** Definition of parameters of the rigid catenary model.

| Parameter | Definition |
|---|---|
| $\rho$ | Density of the simplified beam section |
| $A$ | Cross sectional area of the simplified beam section |
| $E$ | Young's modulus of the simplified beam section |
| $I$ | Moment of inertia of the simplified beam section |
| $L$ | Length of the simplified beam section between supporting points |
| $y$ | Vertical displacement of the simplified beam section |
| $F_c$ | Contact force between the pantograph strip and contact wire |
| $m_c g$ | Force on the supporting arm at the supporting point |
| $m_z g$ | Force on the simplified section at the supporting point |
| $x_0$ | Initial position at the sliding interface between the pantograph strip and contact wire |
| $t_0$ | Initial timestep at the sliding interface between the pantograph strip and contact wire |
| $q_k$ | Cartesian coordinate |
| $M$ | Mass of the simplified beam section |
| $C$ | Damping coefficient of the simplified beam section |
| $K$ | Stiffness of the simplified beam section |

## 2.2. The Pantograph Model

As a preliminary study for the vibration characteristics of the coupled system, the pantograph model adopted in this computational simulation is a lumped mass model. As shown in Figure 6, the simplified model consists of three levels of mass blocks and spring–dashpots, which represent the pan-head, the upper arm and the lower arm, respectively. The equation of motion of the pantograph can be written as Equations (8)–(10) with the parameters defined in Table 2.

$$m_1 \ddot{y}_1 + c_1 \dot{y}_1 + k_1 y_1 - c_1 \dot{y}_2 - k_1 y_2 = -F_c(t) \tag{8}$$

$$m_2 \ddot{y}_2 + (c_1 + c_2) \dot{y}_2 + (k_1 + k_2) y_2 - c_1 \dot{y}_1 - c_2 \dot{y}_3 - k_1 y_1 - k_2 y_3 = 0 \tag{9}$$

$$m_3 \ddot{y}_3 + (c_2 + c_3) \dot{y}_3 + (k_2 + k_3) y_3 - c_2 \dot{y}_2 - k_2 y_2 = F_0 \tag{10}$$

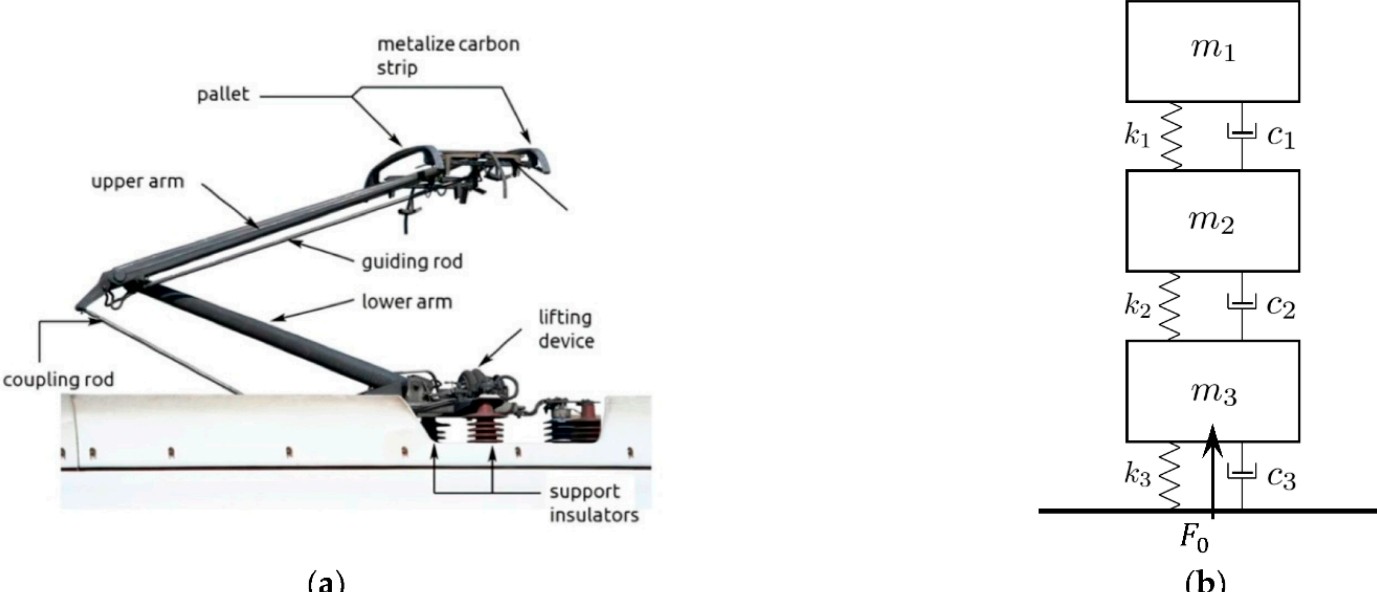

**Figure 6.** A lumped mass model of the pantograph. (**a**) Pantograph and its key components. (**b**) Equivalent lumped mass model.

**Table 2.** Definition of parameters of the pantograph model.

| Parameter | Definition |
|---|---|
| $y_1$ | Vertical displacement of the pan-head |
| $y_2$ | Vertical displacement of the upper arm |
| $y_3$ | Vertical displacement of the lower arm |
| $m_1$ | Mass of the pan-head |
| $m_2$ | Mass of the upper arm |
| $m_3$ | Mass of the lower arm |
| $c_1$ | Damping coefficient between the pan-head and the upper arm |
| $c_2$ | Damping coefficient between the upper arm and the lower arm |
| $c_3$ | Damping coefficient between the lower arm and the roof of the rail vehicle |
| $k_1$ | Stiffness between the pan-head and the upper arm |
| $k_2$ | Stiffness between the upper arm and the lower arm |
| $k_3$ | Stiffness between the lower arm and the roof of the rail vehicle |
| $F_0$ | Static uplift force |
| $F_c$ | Contact force between the pantograph strip and contact wire |

### 2.3. The Pantograph–Catenary Contact Model

The interactive model of the pantograph and rigid catenary system at the sliding interface can be described as shown in Figure 7, with the parameters defined in Table 3. When the contact line is in elastic contact with the pantograph strip, the contact area can be simplified as a rectangle, as per the Hertz theory (Equation (11)). The contact stress at the middle of the contact area is the largest, as shown in Equation (12). Normal and principal shear stress on the symmetrical plane along the $z$-axis of the contact wire is shown in Equations (13) and (14). The half of the contact width at the sliding interface is written as Equation (15).

$$p_x = \frac{2F}{\pi a^2 L} \sqrt{a^2 - x^2} \tag{11}$$

$$p_0 = \frac{2F}{\pi a L} \sqrt{\frac{FE^*}{\pi R L}}; \quad E^* = \frac{E}{1 - v^2} \tag{12}$$

$$\sigma_x = -\frac{p_0}{a}\left(\frac{a^2+2z^2}{\sqrt{a^2+z^2}}-2z\right); \quad \sigma_z = -\frac{p_0 a}{\sqrt{a^2+z^2}}; \quad \sigma_y = v(\sigma_x+\sigma_z) = 2vp_0\left(\frac{z-\sqrt{a^2+z^2}}{a}\right) \tag{13}$$

$$\tau_{xy} = \frac{p_0}{a}\left(z-\frac{z^2}{\sqrt{a^2-z^2}}\right); \quad \tau_{yz} = \tau_{zx} = 0 \tag{14}$$

$$a = \sqrt{\frac{4FR}{\pi E^* L}} = \frac{2p_0 R}{E^*} \tag{15}$$

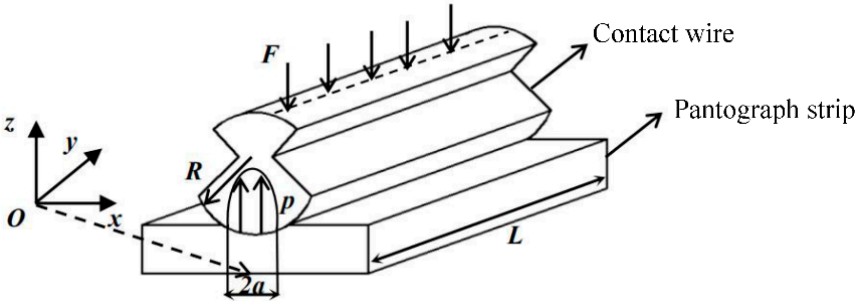

**Figure 7.** The pantograph–catenary contact model.

**Table 3.** Definition of parameters of the pantograph–catenary contact model.

| Parameter | Definition |
|---|---|
| $L$ | Length of the simplified beam section between supporting points |
| $2a$ | Width of the contact area between the pantograph strip and contact wire |
| $z$ | Distance along the $z$-axis from the origin of the Cartesian coordinate |
| $p$ | Distribution of the contact stress |
| $F$ | External load |
| $x$ | Radial distance from the contact point to the middle of the contact area |
| $E$ | Young's Modulus |
| $E^*$ | Equivalent Young's Modulus |
| $v$ | Poisson's ratio |
| $R$ | Radius of the cylinder of the contact wire |
| $\sigma$ | Normal stress on contact wire |
| $\tau$ | Shear stress on contact wire |

### 2.4. The Pantograph–Catenary Coupled Model

A numerical model of the coupled pantograph–catenary system, shown in Figure 8, was established to analyze its vibration response. Model inputs are shown in Tables 4 and 5.

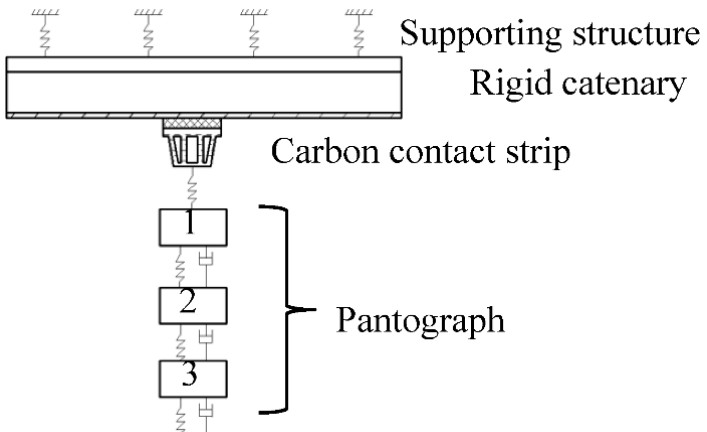

**Figure 8.** An illustration of the coupled pantograph–catenary model.

**Table 4.** Parameter input of the rigid catenary and the pantograph strip.

| | Components | Material | Young's Modulus (GPa) | Density (kg/m³) | Poisson's Ratio |
|---|---|---|---|---|---|
| Rigid catenary | Aluminium profile | Aluminium alloy | 72.0 | 2800 | 0.33 |
| | Contact Line | Copper silver alloy | 120 | 9183 | 0.3 |
| Pantograph strip | Support | Aluminium alloy | 72.0 | 2800 | 0.33 |
| | Carbon strip | Carbon | 12.7 | 2000 | 0.35 |

**Table 5.** Parameter input of the pantograph.

| Mass Block No. | m/kg | k/(N/m) | c/(N.s/m) |
|---|---|---|---|
| 1 | 11 | 13,300 | 0 |
| 2 | 10 | 7540 | 0 |
| 3 | 12 | 3500 | 120 |

A sensitivity study on the appropriate length of the catenary section L was undertaken for the shortest section length, which did not affect the simulation output, and for the verification of the numerical model. A harmonic load of 120 N was applied vertically to the rigid catenary model with different lengths (Figure 9). The vibration response of the model, as per Figure 10, shows that little difference was found for section length from 18 m and above, in terms of peak response and corresponding modal frequencies. Thus, the catenary model with a section length of 18 m between supporting points was chosen for subsequent simulations.

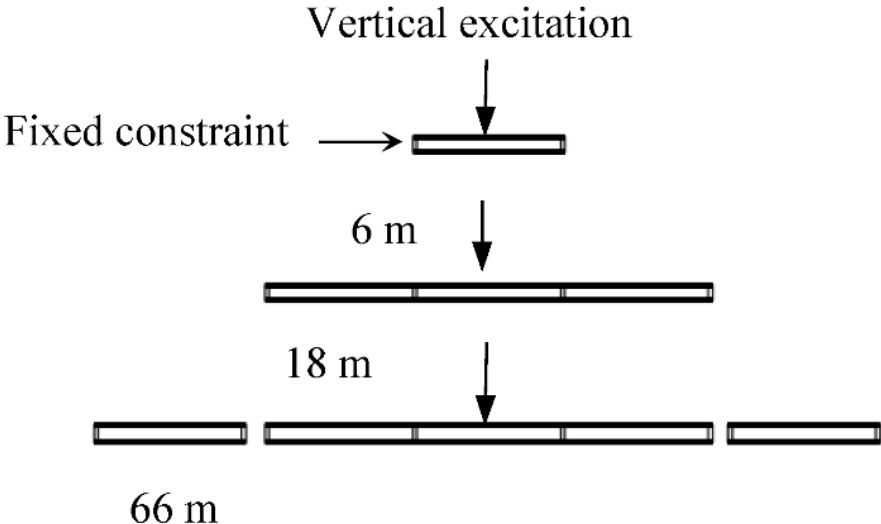

**Figure 9.** Loading case of the rigid catenary model with different section lengths.

*2.5. Simulation Results*

From the computational simulation, modal frequencies and mode shapes of the pantograph frame and those of the pantograph strip are plotted in Figures 11 and 12, respectively. Pantograph strips move together with the pan-head; attributed to the low stiffness of the pantograph frame, in general, the peak acceleration magnitude in the low-frequency range of the pantograph strips is almost negligible compared to the peak acceleration magnitude in the high-frequency range. Therefore, mode shapes corresponding to the free motion of pantograph strips in the low-frequency range are not included herein.

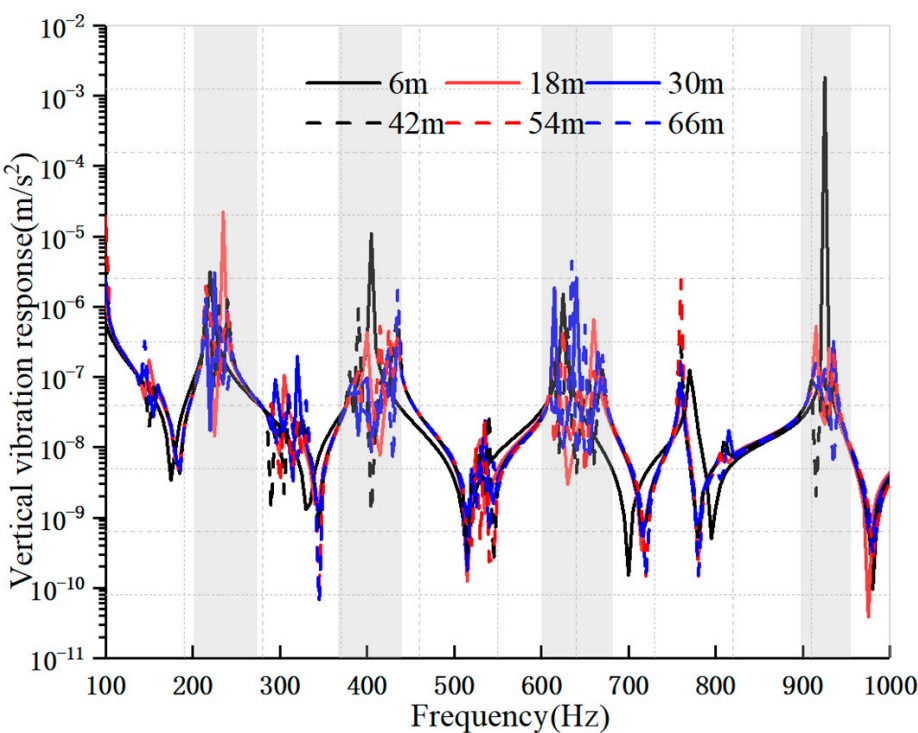

**Figure 10.** Vertical vibration response of the rigid catenary model with different section lengths.

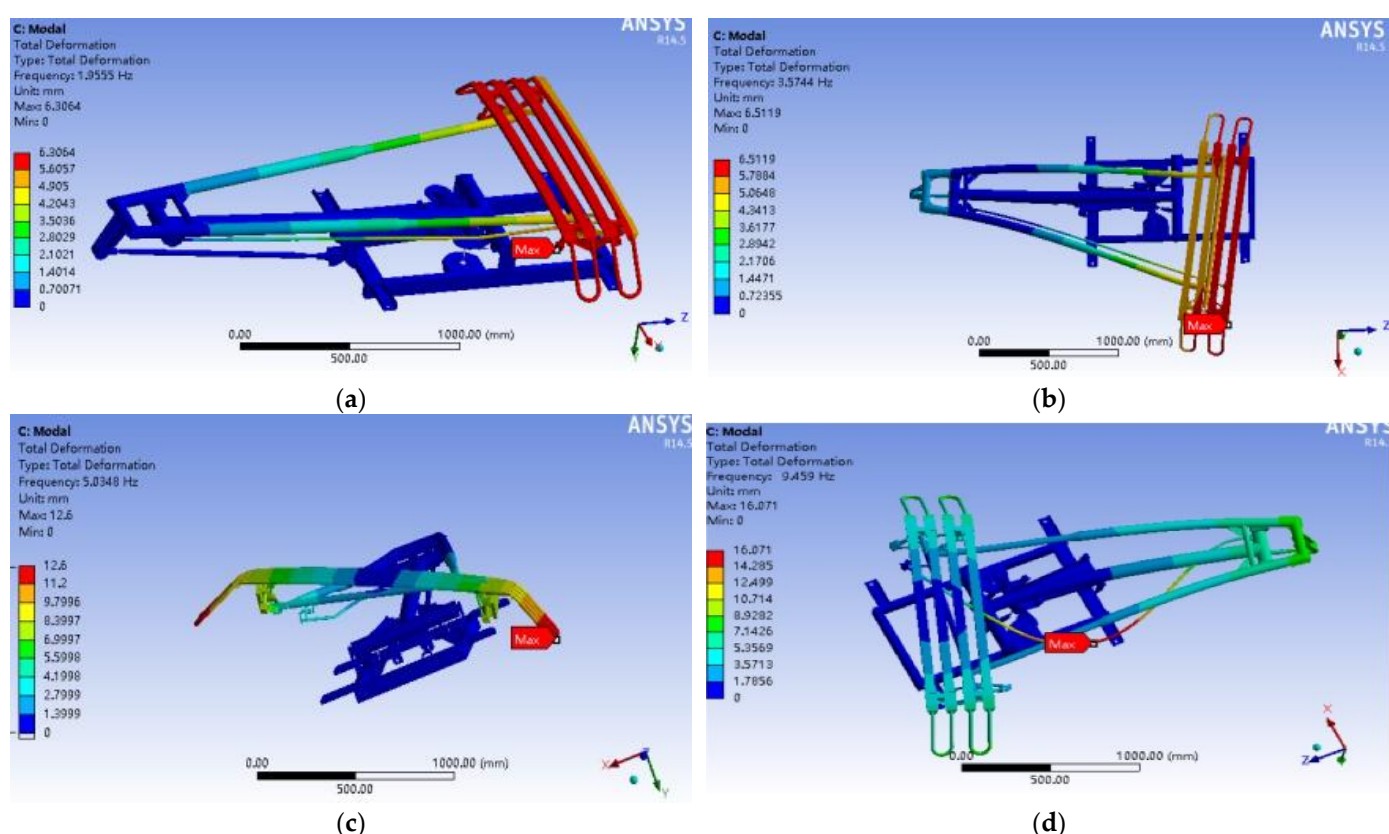

**Figure 11.** Modal frequencies and mode shapes of a pantograph frame. (**a**) First mode, 2.0 Hz. (**b**) Second mode, 3.6 Hz. (**c**) Third mode, 5.1 Hz. (**d**) Fourth mode, 9.4 Hz.

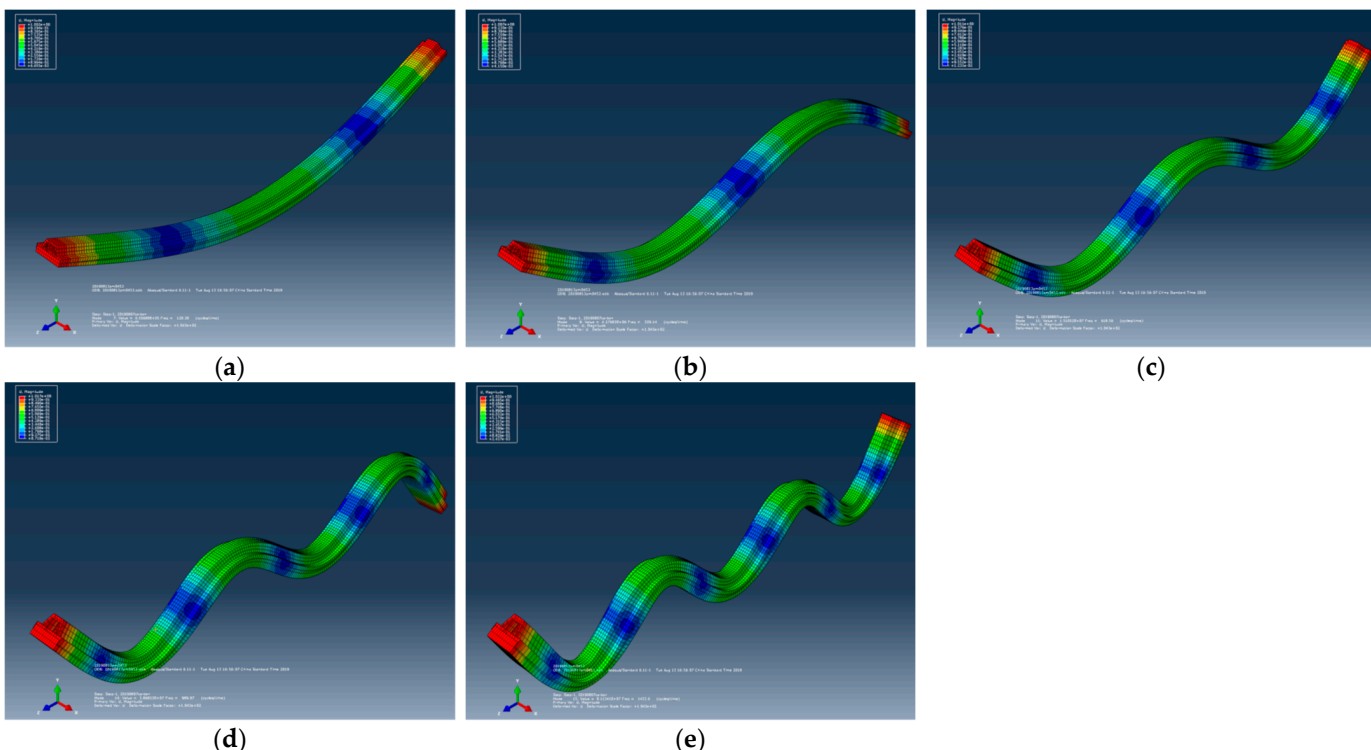

**Figure 12.** Modal frequencies and mode shapes of pantograph strips (from simulation). (**a**) First vertical mode, 116 Hz. (**b**) Second vertical mode, 339 Hz. (**c**) Third vertical mode, 653 Hz. (**d**) Fourth vertical mode, 1005 Hz. (**e**) Fifth vertical mode, 1431 Hz.

## 3. Laboratory Tests

*Test Set-Up*

A modal analysis was carried out on pantograph strips in the laboratory to investigate the vibration characteristics of pantograph strips. As shown in Figure 13, with pantograph strips in free and mounted conditions, two accelerometers measuring vertical and transverse vibration were located at point 4 and 9 along the pantograph strip. A point load was applied at 13 excitation points along the pantograph strip.

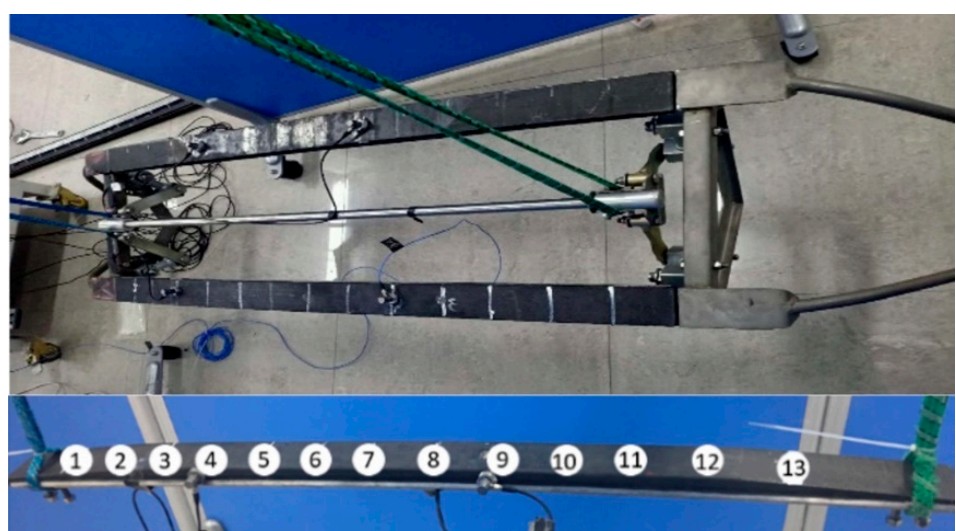

**Figure 13.** Modal test set-up for pantograph strips.

Modal frequencies of the first five vertical modes from the laboratory test are summarized below in Table 6, together with the modal frequencies obtained from the simulation. The modal frequencies from both numerical simulation and from the laboratory test agreed well, with a maximum error of 4.1%. The mode shapes of the five modes from the laboratory tests were also consistent with the simulation outputs. The numerical model was, therefore, successfully validated against the laboratory results.

**Table 6.** Modal frequencies of the pantograph strip.

| Vibration Mode | Simulation (Hz) | Laboratory (Hz) | % Error |
| --- | --- | --- | --- |
| 1 | 116 | 121 | 4.1% |
| 2 | 339 | 334 | 1.5% |
| 3 | 653 | 659 | 0.9% |
| 4 | 1005 | 1016 | 1.1% |
| 5 | 1431 | 1450 | 1.3% |

The frequency response function of the pantograph strip tested is plotted in Figure 14 below. Peak accelerations happened at the second and the third vertical mode, with acceleration magnitudes of 139.6 dB and 148.5 dB, respectively.

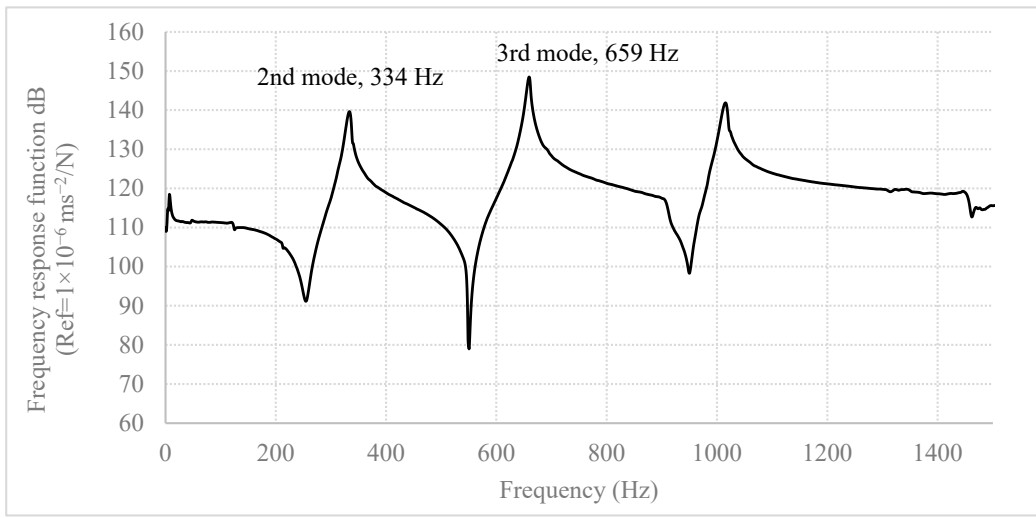

**Figure 14.** Frequency response function of the pantograph strip.

It was found that the wear pattern of the pantograph strips with two-groove wear closely resembled the mode shape of the second vertical mode of the pantograph strip, as shown in Figure 15. Likewise, the three-groove wear pattern was similar to the mode shape of the third vertical mode of the pantograph strip. The higher the acceleration magnitude, the greater the dynamic contact force and thus the greater the friction force, leading to greater wear. The position of the peak acceleration magnitude along the strip corresponds to the deepest wear position. It was also deduced that the irregular wear of the pantograph–catenary system was due to the resonance at the pantograph strip and catenary wire sliding interface in the high-frequency range.

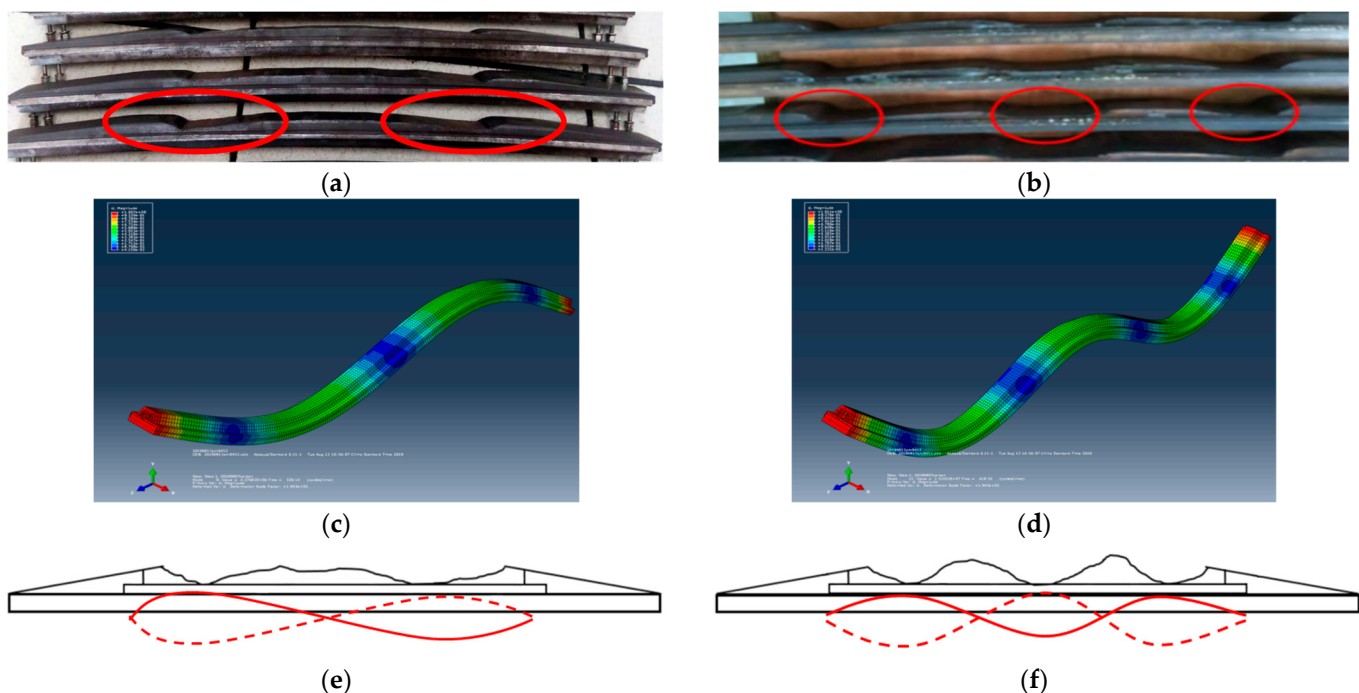

**Figure 15.** Relationship between the wear pattern and the dominant mode shapes of the pantograph strip. (**a**) Irregular wear with two grooves. (**b**) Irregular wear with three grooves. (**c**) Mode shape of the 2nd vertical mode. (**d**) Mode shape of the 3rd vertical mode. (**e**) Two-grooves wear pattern vs. mode shape of the 2nd vertical mode. (**f**) Three-groove wear pattern vs. mode shape of the 3rd vertical mode.

### 4. Improvement in the Damping Property of the Pantograph Strip

Improvement in damping property is one of the most effective means for a system undergoing resonance. A parametric study of the damping property of the pantograph strip was carried out using the numerical model in Section 2. Based on the preliminary study, three types of damping treatment, namely, damping schemes A, B, and C were ap-plied to standard pantograph strips for the control of high-frequency vibration at the sliding contact. Damping ratios at the second and the third vertical mode of pantograph strips using different treatments were compared against those of the original pantograph strip shown in Table 7. Frequency response functions of the four pantograph strips are plotted in Figure 16. Amongst the three damping options, scheme C shows the best damping performance, giving an increase in damping ratio from 0.39% to 7.7% at the third vertical mode, corresponding to three-groove irregular wear. The modal frequency of the third vertical mode shifted from 659 Hz down to 644 Hz, with a significant vibration reduction of 22 dB. Therefore, the scheme C damping treatment of the pantograph strips was further examined with trains running in the field.

**Table 7.** Comparison of the damping ratio of pantograph strips.

| Vibration Mode | Original | | Scheme A | | Scheme B | | Scheme C | |
|---|---|---|---|---|---|---|---|---|
| | F (Hz) | $\zeta$ | F (Hz) | $\zeta$ | F (Hz) | $\zeta$ | F (Hz) | $\zeta$ |
| 2nd vertical mode | 334 | 1.05% | 315 | 1.69% | 314 | 0.96% | 299 | 1.99% |
| 3rd vertical mode | 659 | 0.39% | 625 | 2.81% | 651 | 4.86% | 644 | 7.70% |

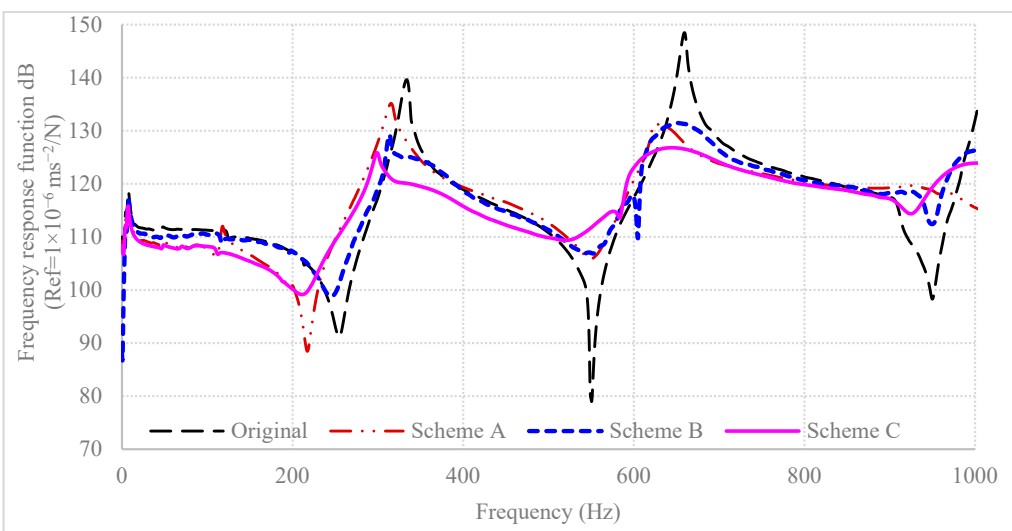

**Figure 16.** Frequency response function of pantograph strips under vertical excitation at point 9 in Figure 13.

To further verify the effectiveness of damping treatment scheme C, field tests were performed between two stops of Metro Line 1 of city X (Figure 17). The test section contained both a traditional catenary with suspension wires and a rigid catenary system, and thus was chosen for the test. The distance between the two stops, stop L and stop X, was 1988 m, with 1044 m of rigid catenary underground and 944 m of traditional catenary on the viaduct. The test route included both straight and curved tracks.

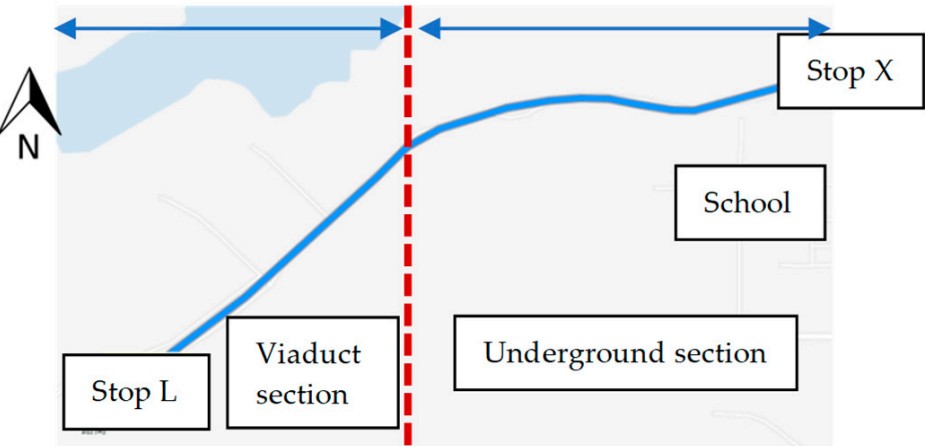

**Figure 17.** Layout of the field test route.

A brand-new pantograph strip without damping treatment, named PS A, and one with scheme C damping treatment, named PS B, were installed on the pantograph of an engineering train. Accelerometers were set up as per Figure 18. Accelerometers 1, 2, 5 and 6 were at the middle of PS A and PS B, respectively. Accelerometers 3, 4, 7, and 8 were at the quarter position of PS A and PS B, respectively. Accelerometers 9 and 10 were at the base of the pantograph frame. Accelerometers 11 and 12 were fixed on the roof of the engineering train. Accelerometers 1, 3, 5, 7, and 9 measured vertical vibration. Accelerometer 2, 4, 6, 8, 10, and 12 measured transverse vibration.

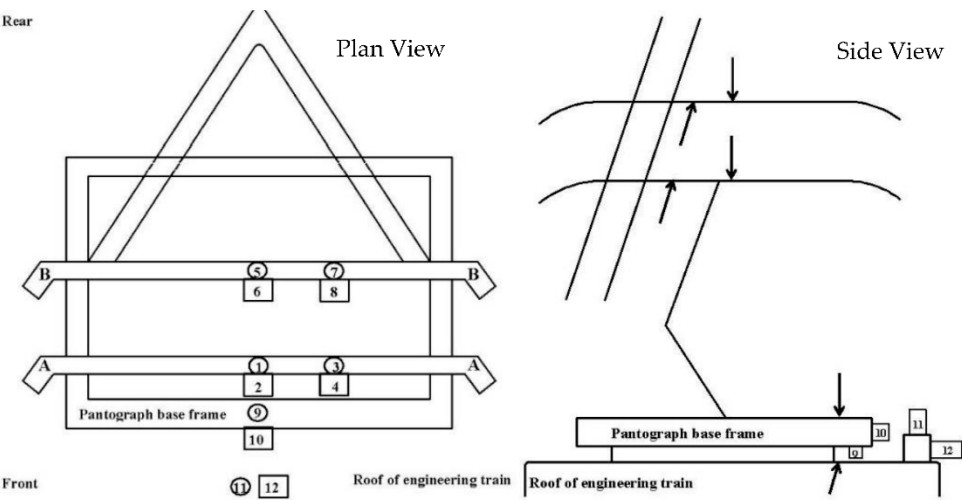

**Figure 18.** Field test set-up.

The new pantograph strip PS A was first tested. Data were collected with the train running at 20 km/h both ways between the two stops, and then at 48 km/h. Damping-treated pantograph strip PS B was then tested under the same conditions. All tests were conducted without the rail vehicle being electrified to examine only the mechanical aspects of irregular wear. All tests were carried out during planned possession time at night for minimum external uncertainties.

Test results in terms of reduction in vibration acceleration comparing the total vibration level from PS A and from PS B are summarized in Table 8. Damping scheme C could achieve up to 7 dB of reduction. In addition, vibration acceleration reduction in the rigid catenary section was generally higher than that found in the traditional catenary section, indicating that the damping effect was more prominent in the rigid catenary system. The higher the train speed, the more reduction in vibration acceleration using damping treatment. In other words, the worse the initial condition of the pantograph–catenary system, the more prominent the damping effects on the control of high-frequency vibration.

**Table 8.** Reduction in vibration acceleration of pantograph strips.

| Train Speed (km/h) | Accelerometer Position | Reduction in Vibration Acceleration (dB) | | | | | | | |
|---|---|---|---|---|---|---|---|---|---|
| | | Rigid Catenary | | | | Traditional Catenary with Suspension Wires | | | |
| | | Upline | | Downline | | Upline | | Downline | |
| | | A | B | A | B | A | B | A | B |
| 20 | Middle | 6.6 | 2.8 | 2.8 | 2.4 | 5.7 | 2.2 | 1.4 | 3.1 |
| | 1/4 position | 3.8 | 2.2 | 5.4 | 1.9 | 1.1 | 2.0 | 2.4 | 1.9 |
| 48 | Middle | 5.9 | 7.0 | 5.5 | 1.5 | 1.9 | 2.0 | 1.0 | 2.6 |
| | 1/4 position | 4.9 | 5.0 | 1.9 | 1.8 | 2.1 | 2.0 | 2.1 | 0.7 |

## 5. Conclusions

This ongoing research work tries to understand the damage mechanism leading to the irregular wear problem in pantograph–catenary coupled systems.

Through computational simulation and laboratory tests, vibration characteristics of the coupled system, especially in the rigid catenary system, were investigated. The following key findings can be concluded:

1. The irregular wear problem is characterized by the consistency between the wear pattern and the mode shape of dominant modal frequencies of pantograph strips, not the pantograph as a whole. Furthermore, this is due to resonance at the pantograph strip and contact wire sliding interface in the high-frequency range;
2. Improvement of damping performance is one of the most effective means to control the high-frequency vibration of the pantograph strip. Damping-treated pantograph strips can reduce vibration acceleration up to 7 dB;
3. By comparing damping performance between traditional and rigid catenary system, and with varying train speeds, it was found that the worse the initial condition of the pantograph–catenary system, the more prominent the damping effects on the control of high-frequency vibration in the irregular wear problem.

**Author Contributions:** Conceptualization, X.P.G., Z.M. and A.W.; methodology, Q.W., X.P.G. and A.W.; software, Q.W. and A.W.; validation, X.P.G. and A.W.; formal analysis, Q.W. and X.P.G.; investigation, Q.W., X.P.G. and A.W.; resources, Z.M. and A.W.; data curation, X.P.G.; writing—original draft preparation, Q.W.; writing—review and editing, X.P.G.; visualization, Q.W.; supervision, X.P.G. and A.W.; project administration, X.P.G.; funding acquisition, A.W. All authors have read and agreed to the published version of the manuscript.

**Funding:** This research received no external funding.

**Institutional Review Board Statement:** Not applicable.

**Informed Consent Statement:** Not applicable.

**Data Availability Statement:** Raw and/or processed data required to reproduce these findings cannot be shared at this time.

**Acknowledgments:** The work was supported by China National Railway Group Co. Ltd. Science, Technology Research and Development Project (L2021G005).

**Conflicts of Interest:** The authors declare no conflict of interest.

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
