# Peer review of "A Study on the Vibration Characteristics and Damage Mechanism of Pantograph Strips in a Railway Electrification System"

_machines, doi:10.3390/machines10080710_

Round 1

Reviewer 1 Report

The scientific publication of the authors "A Study on the Vibration Characteristics and Damage Mechanism of Pantograph Strips in Railway Electrification System" is presented at a high scientific and technical level. The results of both theoretical and experimental studies are presented. At the beginning of the article, a broad analysis of the publication is presented, the problem presented is widely disclosed.

I believe that the issue in the field of railway transport is very relevant. This is related to both traffic safety and operating costs. The problem of wear in contact between a pantograph and a contact network has not been fully studied at present, therefore I consider these studies to be relevant and promising.

The above results are presented and justified to a sufficient extent, so I consider it necessary to accept the article in this form.

Author Response

Dear reviewer,

Thank you very much for your time and support.

Best regards,

Phrain

Reviewer 2 Report

Dear Authors,

The article sent for review entitled: A Study on the Vibration Characteristics and Damage Mechanism of Pantograph Strips in Railway Electrification System presents an interesting approach to the topic presented.

The work consists of 5 chapters, the structure of which is arranged in a logical whole. The mathematical model used for the presented system, as well as the use of the finite element method, deserve special attention and praise.

In the presented structure, the article requires some insignificant substantive corrections:

- Figure 3 is not very legible, perhaps black color should be used instead of gray.

- Quotations shown on line 43, for example, should be separated by a space - see [6, 7].

- Mathematical relationships should be arranged in the text so that they do not go beyond the text area.

- Table 3 lacks the denominators by which the variables are described.

- In Figure 7, the coordinate system should be brought outward, showing its axes, this will be better readable, and the point of its attachment should be indicated in the figure itself.

- In the text, it should be better explained why in Figure 10 the authors indicated peak ranges of 400, 600 and 900, and the peak of 250 was omitted?

Conclusion: 

In order to proceed further with the publication process, the reviewer would like to encourage the authors to comply with the comments/suggestions. A thorough revision of the paper is required. 

Author Response

Dear reviewer,

Thank you very much for your time and advice on our paper. We have made amendments according to each of your suggestion.

Best regards,

Phrain.

Reviewer 3 Report

The problem presented by the authors is not new. Many researchers focus their work on the pantograph-catenary interaction.

The authors use the word “train” for each rail vehicle. I suggest changing it to the “rail vehicle” because rail vehicles include trains, electric multiple units (trainset) and trams. Presented by the authors' the problem is well-known in each of these kinds of rail vehicles.

Is figure 3 present the wear of strip in milometers (the figure does not mention it)?

Why does figure 3 present just Vehicle 2 and Vehicle 5? How many vehicles were analysed? What is presented on the horizontal axis, year and month? There are no 14 months in a year, so it is not clear.

Was the strip thickness presented in figure 3 measured in just one place or it is an average value from a few places?

How many vehicles and what period were used for collecting the data?

The state of the art was omitted. I suggest preparing it, there is much interesting paper, i.e.: Pantograph Sliding Strips Failure—Reliability Assessment and Damage Reduction Method Based on Decision Tree Model, Materials 2021, 14(19); A method of predicting wear and damage of pantograph sliding strips based on Artificial Neural Networks, Materials 2022, and more paper connected.

In the conclusion, the results (values) from the presented research should be mentioned.

What will be the future work related to this area of research?

Author Response

Dear reviewer,

Thank you very much for your time and advice on our paper. We have made amendments according to your suggestions. Below are our response to your questions.

Q1: The problem presented by the authors is not new. Many researchers focus their work on the pantograph-catenary interaction.

R1: Despite the amount of research work done on pantograph-catenary interaction, the condition of the rigid catenary system, especially its impact on the coupled system is less well understood. The aim of this particular research is to look into the irregular wear problem, which has not been found in literatures world-wide.

Q2: The authors use the word “train” for each rail vehicle. I suggest changing it to the “rail vehicle” because rail vehicles include trains, electric multiple units (trainset) and trams. Presented by the authors' the problem is well-known in each of these kinds of rail vehicles.

R2: Changed per suggestion.

Q3: Is figure 3 present the wear of strip in milometers (the figure does not mention it)?

R3: The figure presents the wear of strip in mm per 10,000 km running, which is explained in the main text (line 106 and 107).

Q4: Why does figure 3 present just Vehicle 2 and Vehicle 5? How many vehicles were analysed? What is presented on the horizontal axis, year and month? There are no 14 months in a year, so it is not clear.

R4: The train consists of 6 rail vehicles, two carriages with motor providing traction, and four without motors. So only the 2nd and the 5th rail vehicle have pantograph and pantograph strips connected to overhead lines. Date form changed.

Q5: Was the strip thickness presented in figure 3 measured in just one place or it is an average value from a few places?

R5: Average values measured at troughs, as added in the text (line 104 & 105)

Q6: How many vehicles and what period were used for collecting the data?

R6: The data presented is provided by the rail operator, as a statistical means to present the problem. The number of vehicles measured is not particularly relevant to the analysis and evaluation of the irregular problem. The period is shown in Fig. 3, from 1st Dec to 15th Dec., for half a month.

Q7: The state of the art was omitted. I suggest preparing it, there is much interesting paper, i.e.: Pantograph Sliding Strips Failure—Reliability Assessment and Damage Reduction Method Based on Decision Tree Model, Materials 2021, 14(19); A method of predicting wear and damage of pantograph sliding strips based on Artificial Neural Networks, Materials 2022, and more paper connected.

R7: Traditional research on pantograph catenary interaction, without the irregular wear problem, focuses primarily on electrical wear and low-frequency vibration of the pantograph itself. This research, based on preliminary observations, looks into the irregular wear problem, especially the dynamic vibration characteristics in high-frequency range at the pantograph strip and contact line interface. The authors believe the above research focus is new, and sheds insights to future works in the attempt to solve the irregular wear problem.

Q8: In the conclusion, the results (values) from the presented research should be mentioned.

What will be the future work related to this area of research?

R8: This research work aims to understand better the mechanism of the irregular wear problem, the solution to the problem reduces operational safety hazard, as well as maintenance costs.

Response to ‘must be improved’ questions: All references cited are relevant, in some aspects, to the research. The description of research methods and results have been improved according to the reviewer’s suggestions.

Kind regards,

Phrain.

Round 2

Reviewer 3 Report

The state of the art should be extended. Authors do not do this after correction the paper.

Author Response

Dear reviewer,

Our research, as introduced in the opening section and the literature review (line 72-76) of the paper, looks into the areas which have been overlooked so far, including the irregular wear problem, especially the dynamic vibration characteristics in the high-frequency range at the pantograph strip and contact line interface; whereas existing research mainly studies low frequency vibration of the pantograph. Irregular wear problem is more prominent in the rigid catenary system, there is very limited research work done and reported on the rigid catenary system.

In the next stage of our research, we plan to establish the correlation between vibration reduction and reduction in wear, using computational simulation and a 1:5 scaled full model of the pantograph-catenary train track system. We believe our research outputs valuable towards solving the irregular wear problem which causes operational safety hazard and increases life-cycle costs.

You kindly gave an example of using artificial neural network to predict pantograph-catenary wear for our reference. Artificial neural network is a more recent technology which I myself came across when doing my PhD, but I do not consider it appropriate, nor is it necessary for this particular research work. As the title of our paper states, we try to understand the fundamental mechanism of the irregular wear, and before we get a grip on the mechanism and cause of irregular wear, which we believe is different from normal wear of the coupled system, we find it difficult and unconvincing to predict irregular wear using the newer technology.

So to summarise, we aim at the understanding of the mechanism of the irregular wear problem mainly found in the rigid catenary system. We believe the mechanism of irregular wear is different from that of normal wear. We investigate the dynamic vibration characteristics in the high-frequency range at the pantograph strip and contact line interface, rather than existing research on low frequency vibration of the pantograph. We will be using our 1:5 scaled full model of the pantograph-catenary train track system to examine the relationship between vibration reduction and reduction in wear in the irregular wear problem.

We hope the above alleviates your concern of ‘the-state-of-the-art’.

Thank you again for your time and comments on our paper.

Best regards,

Phrain.
